The impact of curcumin derived polyphenols on the structure and flexibility COVID-19 main protease binding pocket: a molecular dynamics simulation study

http://orcid.org/0000-0001-6705-5338 Mulu Aweke 1 aweke.mulu@aastu.edu.et
Gajaa Mulugeta 2
Woldekidan Haregewoin Bezu 1
W/mariam Jerusalem Fekadu 1
1 College of Applied Science, Addis Ababa Science and Technology University , Addis Ababa , Ethiopia
2 College of Natural and Social science, Addis Ababa Science and Technology University , Addis Ababa , Ethiopia
Orlov Yuriy
Electronic publication date: 2021 Jul 19
Publication date: 2021
Volume: 9
Electronic Location ID: e11590
Received 2020 Jun 5; Accepted 2021 May 20
Copyright: © 2021 et al.
Copyright year: 2021
Copyright holder: et al.
License: This is an open access article distributed under the terms of the Creative Commons Attribution License, which permits unrestricted use, distribution, reproduction and adaptation in any medium and for any purpose provided that it is properly attributed. For attribution, the original author(s), title, publication source (PeerJ) and either DOI or URL of the article must be cited.
License URL: https://creativecommons.org/licenses/by/4.0/

Keywords: NAMD, X-ray crustal structure, SARS-CoV-2, Docking, Binding energy

Funding: University of Addis Ababa Science and Technology This work was supported by the University of Addis Ababa Science and Technology for using computational resources. The funders had no role in study design, data collection and analysis, decision to publish, or preparation of the manuscript.

==============================
The newly occurred SARS-CoV-2 caused a leading pandemic of coronavirus disease (COVID-19). Up to now it has infected more than one hundred sixty million and killed more than three million people according to 14 May 2021 World Health Organization report. So far, different types of studies have been conducted to develop an anti-viral drug for COVID-19 with no success yet. As part of this, silico were studied to discover and introduce COVID-19 antiviral drugs and results showed that protease inhibitors could be very effective in controlling. This study aims to investigate the binding affinity of three curcumin derived polyphenols against COVID-19 the main protease (Mpro), binding pocket, and identification of important residues for interaction. In this study, molecular modeling, auto-dock coupled with molecular dynamics simulations were performed to analyze the conformational, and stability of COVID-19 binding pocket with diferuloylmethane, demethoxycurcumin, and bisdemethoxycurcumin. All three compounds have shown binding affinity −39, −89 and −169.7, respectively. Demethoxycurcumin and bisdemethoxycurcumin showed an optimum binding affinity with target molecule and these could be one of potential ligands for COVID-19 therapy. And also, COVID-19 main protease binding pocket binds with the interface region by one hydrogen bond. Moreover, the MD simulation parameters indicated that demethoxycurcumin and bisdemethoxycurcumin were stable during the simulation run. These findings can be used as a baseline to develop therapeutics with curcumin derived polyphenols against COVID-19.

Introduction

The severe acute respiratory syndrome coronavirus 2 (SARS-CoV-2), which was first reported in Hubei Province of Wuhan, China in December 2019, is responsible for the ongoing global pandemic of coronavirus disease 2019 (COVID-19) spreading across almost all countries with 160,686,749 active infection cases and 3,335,948 deaths until 14 May 2021 (World Health Organization, 2020; Guarner, 2020; WHO, 2020). The actual number of cases presumed much higher, due to limitations in testing, lack of medication, and applying World Health Organization recommendations such as social distancing, hand washing, and travel ban and full lockdown in many cities (World Health Organization, 2020; Chen, Liu & Guo, 2020; WHO, 2020). Prior work in this area showed that SARS-CoV-2 is the third coronavirus belonging to Genus Beta coronavirus that can infect human next coronavirus-severe acute respiratory syndrome (SARS-CoV-2) and the Middle East respiratory syndrome (MERS-CoV) (Guarner, 2020). SARS-CoV-2, the virus responsible for COVID-19, belongs to a group of genetically-related viruses that includes SARS-COV and some other CoVs isolated from bat populations (World Health Organization, 2020). The SARS-CoV-2 virus is the primary causative agent of COVID-19. The pol gene of the SARS-CoV-2 virus possesses a positive-sense ∼30 kb long RNA genome with ~14 ORFs and encoding 27 proteins (Chen, Liu & Guo, 2020; Hoque et al., 2020). Among multiple encoded proteins programmed ribosomal frameshifting generates two polyproteins encoding the replicase proteins (Zhao, Weber & Yang, 2013). The range of viral processing protein associated with structural proteins, non-structural proteins, and accessory proteins have been reported in several studies (World Health Organization, 2020; Zhao, Weber & Yang, 2013; Pal et al., 2020). The pol gene of the SARS-CoV-2 virus encodes two protease enzymes main protease (Mpro) and Papain-like protease, which are involved in the proteolytic processing of the polyproteins into individual nsp7/nsp8/nsp12 to control viral gene expression and replication in the host (Chen, Liu & Guo, 2020; Zhao, Weber & Yang, 2013; Pal et al., 2020). SARS-CoV-2 virus Mpro is a vital enzyme that catalyzes the proteolytic process, non-structural proteins (nsps) generated by the main protease play a major role in the reverse-transcribed viral DNA into the host genome (Zhao, Weber & Yang, 2013; Woo et al., 2010; Elaswad et al., 2020; Parlikar et al., 2020). It is composed of three structurally and functionally distinct domains I, II, and III. Domain I and II contain highly conserved Cys145-His41 residues located in the cleft which are directly involved in the catalytic activities of Mpro (Jin et al., 2020). Therefore, SARS-CoV-2 virus Mpro has emerged as a promising antiviral target as this is responsible for newly synthesized double-stranded viral DNA and subsequent maturation of polyproteins into host genomic DNA. SARS-CoV-2 virus Mpro inhibitors repurposing, which target the enzyme active site, have been studied computationally over the past few months to control COVID-19 as there is an urgent requirement for a strong drug or combination of drugs to combat the pandemic (Muralidharan et al., 2020; Liu et al., 2020). Currently, natural products are screened by molecular docking and molecular dynamic simulation to test their affinity towards molecular targets of COVID-19 taking the advantage that natural products are free from toxic or side effects (Elaswad et al., 2020; Emran et al., 2015; Dash et al., 2014). In this study, the structure of SARS-CoV-2 virus Mpro was obtained from the protein data bank (PDB) gave a major advance for structure-based drug design of Curcumin derivative polyphenols, and gives well-established techniques to reveal important dynamic information in proteins. This is an exciting new approach to develop effective therapies to fight COVID-19 with Curcumin, exploiting a new mechanism based on well-validated virulence factors. And also, expanding the list of lead compounds means more possible drugs that could be advanced through clinical trials and better treatments will be possible shortly.

Up to now, there are no robust drugs for the widespread SARS-CoV-2 virus although there are now several vaccines that are in use. Finding a new drug in a wet lab and bringing it to market is hard, expensive, and time-consuming. Due to rapid ongoing global health emergency in the current outbreak and high mortality rate estimated by World Health Organization, the more rapid development of new antiviral drugs is highly demanded (WHO, 2020; Hoque et al., 2020). In the literature, traditional medicines have been extensively investigated to find novel therapeutic strategies for viruses including SARS-CoV-2. Evidence from several computational studies indicated that natural compounds for virus infection treatment play a critical role. A review of the literature showed that natural compounds with low cytotoxicity and high bioavailability seem to be the most efficient candidate’s therapy. Also, studies observed lining modern medicine, humans have relayed to the use of phytochemicals for the treatment of different diseases (Islam, Khan & Mishra, 2019; Forni et al., 2019). In general, work to date in this area supports phytochemicals such as polyphenols possessing a variety of potential biological benefits such as anti-oxidant, anti-inflammatory, antiviral, and antibacterial benefits (Hatcher et al., 2008; Anbarasu & Jayanthi, 2018). A systematic review of peer-reviewed literature showed that Curcumin has antiviral activity in human immunodeficiency virus, herpes simplex virus, dengue virus, Zika, and chikungunya viruses (Hatcher et al., 2008; Anbarasu & Jayanthi, 2018; Mounce et al., 2017; Praditya et al., 2019a). The main objective of this work is to investigate methods for improving Curcumin derivative polyphenols that will be required for developing Mpro inhibitors were performed with silico study that included molecular docking combined with molecular dynamic simulation depend on Mpro binding pocket (PDB ID: 7BUY) (Jin et al., 2020). Specifically, we aim to investigate the amino acids that contribute the most to the binding Curcumin-derived polyphenols; we examined the binding mode of the Mpro pocket and found that form hydrogen bonds with binding pocket of residues. Finally, we put into MD simulation and binding free energy calculation. Results revealed that demethoxycurcumin and bisdemethoxycurcumin (especially bisdemethoxycurcumin) could have good inhibitory activity towards Mpro.

Materials & methods

Protein and polyphenols structure preparation

The crystal structure of the MPro: carmofur complex (PDB ID: 7BUY) (Jin et al., 2020) was retrieved from the Protein Data Bank (PDB). A high-quality homology model of Apo-enzyme (without carmofur) was calculated using MODELLER version 9.10 (Vanommeslaeghe, Raman & MacKerell, 2012). Before docking, crystallographic water molecules and bound ligands were removed from 3D structures. The protonation states of acidic and basic residues were determined under pH 7.0 conditions and analyzed by the H++ server (Anandakrishnan, Aguilar & Onufriev, 2012). Next, the Modell structure was subjected to Discovery studio for optimization and minimization. Finally, validate the modeled structure using the UCLA-DOE server (http://servicesn.mbi.ucla.edu/) (Benkert, Künzli & Schwede, 2009; Colovos & Yeates, 1993; Pontius, Richelle & Wodak, 1996). The modeled 3D structure was then validated and confirmed by using the RAMPAGE, ERRAT, and Verify3D online servers. After validating the homology models, we resolved the issue of mismatched residue and missed structure across the two models. The SMILES of the three natural polyphenols diferuloylmethane, demethoxycurcumin, and bisdemethoxycurcumin were obtained from Pub Chem (Bolton et al., 2008) and their accession numbers are DB969516, DB5469424, and DB5315472, respectively. These SMILES were converted to PDB format with 3 D coordinates using Open Babel (O’Boyle et al., 2011), an open-source chemical toolbox for the inter-conversion of chemical structures.

Docking methodology

The experiments were performed with Auto-Dock 4.2 according to the previous study procedure (Morris et al., 2009). The work was separated into two parts; the first was prepared ligand and protein pdbqt file and Docking Parameter file using Auto-Dock 4.2. The second was performed molecular docking and finally, the results were analyzed. Fourteen refined 3D polyphenols structures were screened for binding affinity and selectivity toward protein. To cover the whole protein structure global docking was conducted with the spacing of 0.5 Å. Subsequently, rigid or flexible docking of the target was performed and then the complexes with the lowest binding energies are selected. The genetic algorithm was used to evaluate parameters with the default setting, and Lamarckian GA (4.2) was employed for docking simulations. All docking parameters procedures were set to 150 genetic algorithm runs using the Lamarckian genetic algorithm conformational search, with the population size of 300, 2,500,000 maximum numbers of energy evaluations, and 27,000 generations per run. The best protein-polyphenols complexes were selected according to the molecular docking results including binding energy, root mean square deviation, and type of favorable interactions and binding sites.

Set simulation parameters and molecular dynamics

To explore the conformational flexibility of best hit compounds against free protein, an MD simulation study was employed using NAMD 2.3 tool on the Windows Operating System (Nelson et al., 1996). Each complex was separated for generating topology and coordinates. The protein topology and coordinate files were lacking in the Amber force field prepared by the general amber force field (CgenFF) Server (Vanommeslaeghe, Raman & MacKerell, 2012), while the ligands topology and coordinate files were generated using Open Babel (O’Boyle et al., 2011). The generated NAMD compatible files for the proteins and the ligands were then merged, and then complex subjected to solvated, minimized, and equilibrated. The systems were solvated in a cubic water-box with the explicit solvation model TIP3P.20 we used a distance of 1.0 Å between the cell wall and the solvated atoms of the system. Counter-ions were also added to neutralize the system. The energy minimization (n steps = 5,000) was conducted using the steepest descent approach (1,000 ps) for each protein-ligand complex. Particle Mesh Ewald (PME) method was employed for energy calculation and electrostatic and Van der Waals interactions; cut-off distance for the short-range Van der Waals was set to 10 Å, where Coulomb cut-off and neighbor list were fixed at 8 Å. Finally, a 50 ns molecular dynamics simulation was carried out for all the complexes with n steps 1,000,000. The RMSD, hydrogen bond distribution, and RMSF analysis were carried out using MS Excel (2016), VMD, and UCSF Chimera 1.10.1 software. Trajectory snapshots were stored at every 0.2 ps during the simulation period, and 3D coordinate files were harvested after every 2 fs for post-dynamic analysis.

MM-PBSA approach Interaction energy

The binding energy of Mpro-Curcumin derivative polyphenols complexes was calculated using the molecular mechanics Poisson–Boltzmann surface area (MM-PBSA) method (Kumari, Kumari & Lynn, 2014). Free energy of solvation (TP3 solvent model) and molecular mechanics potential energy was calculated. In this study, snapshots were taken from MD simulation used for calculating binding free energy. The individual contributions of protein residues to the three energetic components were determined through per-residue decomposition. The temperature used for Poisson–Boltzmann calculation was set to 300 K.

Prediction of ADME by computational analysis

ADME analysis of Bisdemethoxycurcumin, Demethoxycurcumin, and Diferuloylmethane at pH 7 was examined using online software tools (Baell & Holloway, 2012). The important parameters related to ADME properties such as Proudfoot’s rule of five, pharmacokinetic properties, drug likeliness, molar refractivity, and solubility of the drug was deliberated (Daina, Michielin & Zoete, 2017).

Discussion

Until now, no specific antiviral medication was discovered for COVID-19 the only control measure includes proper hand washing, social distancing, mouth and nose masking, travel restriction, and infected person isolation (World Health Organization, 2020; Guarner, 2020; Zhao, Weber & Yang, 2013). Furthermore, Scientists around the world are developing many potential vaccines for COVID-19 and the biggest vaccination campaign in history is underway. However, according to the world meter report, the coronavirus COVID-19 is still affecting 219 countries and worldwide 160,686,749 people infected and 3,335,948 people died until May 14, 2021, 20:04 GMT. In the context of Ethiopia, this virus died about 3,951 people and infected 264,960 people. Viruses continually change through mutations and the gene encoding the S protein of SARS-CoV-2, various mutations have been reported and recently, the United Kingdom (UK) (VOC-202012/01 or VUI-202012/01 or B.1.1.7), Indian (B.1.617), and South Africa (501Y.V2 or 20C/501Y.V2B.1.351) has faced a rapid increase in COVID-19 mediated by new variants (Dawood, 2020; Korber et al., 2020). One way of recovering from this problem could be to identify potent COVID-19 inhibitors are urgently needed, as the situation getting worst. In this regard, plants serve as the source of therapeutic ingredients in the history of the human race. As far as we know, little previous research has investigated curcumin sp polyphenols inhibitory effect on COVID-19. Curcumin consists of three major compounds, namely, diferuloylmethane (60–70%), demethoxycurcumin (20–27%), and Bisdemethoxycurcumin (10–15%) (Heger et al., 2013; Nelson et al., 2017). The review showed Curcumin-derived polyphenols exhibit antiviral activity for influenza, hepatitis C virus, HIV, anti-inflammatory, antibacterial, and antioxidant properties and also exert substantial anticarcinogenic activities (Heger et al., 2013; Nelson et al., 2017; Praditya et al., 2019b). Wet lab experimental approaches for the study of interactions between therapeutic substance and target proteins are not cheap, not safe, labors, and time-consuming (Basak et al., 2020; Bachmann & Lewis, 2005). There are many alternative methods are available for solving these problems. In this study, we proposed the combinational strategy of docking and molecular dynamics simulation. A series of recent studies have indicated that combinational strategy of the docking and molecular dynamic simulations techniques identify the targets for several bioactive compounds whose in vivo targets are unknown including COVID-19 (Rahman et al., 2020; Fiesco-Sepúlveda & Serrano-Bermúdez, 2020; Cortés-García et al., 2020). This study aims to investigate the binding affinity of three Curcumin-derived polyphenols against COVID-19 the main protease (Mpro) binding pocket (Mpro) and the identification of important residues for interaction with curcumin-derived polyphenols. For this purpose, we used protein-carmofur (PDB code: 7BUY) as a model to offer evidence that could elucidate the mechanisms of curcumin sp polyphenols inhibition. At the beginning of this study, curcumin sp-derived polyphenols were subjected to ADMETox evaluation. According to Proudfooti’s rule states that for any compound to be selected as a potential drug it should have; molar refractivity between 40 and 130, Less than 10 hydrogen bond acceptors, less than 5 hydrogen bond donors, high lipophilicity (expressed as LogP ≤ 5) and Molecular mass <500 Dalton. Therefore, three selected curcumin sp-derived polyphenols used in this study passed all the five criteria mentioned in Proudfoot’s rule (Table S3). Demethoxycurcumin and bisdemethoxycurcumin polyphenol passed the ADME evaluation and throughout the screening, only Diferuloylmethane passed the predicted toxicity evaluation. Thus this study suggests that these phytochemicals have the potential to work effectively as drugs. Next, to validate our docking methodology crystal structure of COVID-19 main protease binding pocket (Mpro) (PDB code: 7BUY) with carmofur bound at the binding pocket was retrieved from PDB and re-dock with carmofur. The values of the RMSD between the Mpro crystals structure (RMSD 1.6) and the re-dock structures (RMSD 1.9) confirmed a close-match between docked and original structures. This confirmed our model structure is good quality and continued docking our work. Molecular docking studies showed that curcumin polyphenols have binding energy values between −12.17 to −12.32 kcal/mol−1 as shown in Table 1. Recent theoretical developments have revealed that Docking studies on different traditional medicine inhibitors activity for the SARS-CoV-2 protease. Several studies have been reported the binding energy of molecules less than or/and higher values compared to this study. Cherrak et al. reported in the literature that Quercetin-3-O-rhamnoside showed the highest binding affinity −7kcal/mol−1 (Cherrak, Merzouk & Mokhtari-Soulimane, 2020). Moreover, it was observed that curcumin exhibited the highest binding free energy of −18.21 kcal/mol−1 in different COVID-19 main protease, which is targeted in this study (Adeoye, 2020). And also, docking of SARS-CoV-2 spike protein (PDB ID: 6LU7) with nelfinavir, lopinavir, kaempferol, quercetin, luteolin-7-glucoside,demethoxycurcumin, naringenin, apigenin-7-glucoside, oleuropein, curcumin, catechin, epicatechin-gallate, gingerol, gingerol, and allicin binding energies ranges between −4.03 to −7.6 kcal/mol−1 (Khaerunnisa et al., 2020). Recently, Computational docking of 14 compounds representing flavonoids, phenolic acids, and terpenes from honey and propolis with two different targets from COVID-19 showed binding affinity ranges between −5.6 to −7.8 kcal/mol−1 (Dawood, 2020). Bisdemethoxycurcumin, Demethoxycurcumin, and Diferuloylmethane revealed a minimum of two hydrogen bonds and a maximum of three hydrogen bonds for the Mpro binding pocket of SARS-CoV-2 (Table 1). In line with previous studies, hydrogen bonds are the most important bonds in determining the binding affinity, selectivity, and stabilization effect of curcumin-derived polyphenols with Mpro (Baby et al., 2020). The inhibition constant (Ki) is higher for Demethoxycurcumin (281.22 uM) than Bisdemethoxycurcumin (98.83 uM) and Diferuloylmethane (156.42 uM) as shown in Table 1. Prior research suggests that the smaller the Ki, the greater the binding affinity and the smaller amount of medication needed to inhibit the activity of that enzyme (Bachmann & Lewis, 2005). The RMSD values obtained for the lowest-energy poses predicted for each polyphenol are shown in Table 1 and average RMSD results ranging from 57.0 to 70.0 Å RMSD. But this is not relevant data for this study.

Table 1 Molecular docking results to determine the SARS CoV-2 main protease (Mpro) binding pocket affinity towards selected curcumin sp derived polyphenols at 298.15 K.

Protein-ligand complexes	Binding energy (kcal/mol)	Inhibition constant (Ki) (uM)	RMSDb (A0)	Residues involved in interaction	No of hydrogen bonds	
Diferuloylmethane-COVID-19 main protease	−12.25	156.42	70.645	Asn142 and Gln192	2	
Demethoxycurcumin-COVID-19 main protease	−12.17	281.22	57.842	Leu272, Thr199,Lys137	3	
Bisdemethoxycurcumin-COVID-19 main protease	−12.32	98.83	65.030	Phe294, Gln110 and Glu240	3	

Furthermore, residues forming stable contact with polyphenols include Phe294, Gln110, Gln240, Gln192 Asn142, Lys137, Thr199, and Leu272 with probabilities larger than 85% (Fig. 5, Table 1). Bisdemethoxycurcumin and Demethoxycurcumin have lost hydrophobic interaction with Pro293 with Tyr239, respectively (Fig. 1). Besides this, a total of eight residue pairs that have average distances smaller than 10.0 Å within free enzyme residues and Mpro-polyphenols complexes binding pocket residues and surface residues were reported (Table 2). This implies that residue from both surface and binding pocket protein is involved in direct contact with curcumin-derived polyphenols. Furthermore, the inter-residue distances between the binding pocket and surface residues in Mpro-bisdemethoxycurcumin and Mpro-demethoxycurcumin complex are slightly different from that of Mpro-diferuloylmethane, particularly for Pro252/Lys137, Thr292/Thr199, Ile200/Leu272, and Arg188/Phe294 residues (Table 2). In line with previous studies, residue serine interacts with residue Phe140 and Glu166 to stabilize the enzyme- polyphenols complex binding site (Jin et al., 2020). For residues Gln192, Leu167, Ser144, Gly143, His114, Pro168, His41, and Cys145, their contact probabilities with bisdemethoxycurcumin were increased significantly and Cys145 is a key catalytic residue which is consistent with a previous study (Jin et al., 2020). Residues forming stable contact with polyphenol include Phe294, Gln110, Gln240, Gln192 Asn142 Lys137 Thr199, and Leu272 the complexes with probabilities larger than 85% (Fig. 1). Also, for residues Glu288, Asp289, Leu287 Arg131, Tyr239, Leu271, Asn274, Gly275, and Leu286 the probabilities of contacts between demethoxycurcumin increased significantly after binding and they also form stable contacts (Fig. 2). For residues Arg188, Val202, Gly109, Pro293, Ile249, Pro252, Thr292, Ile200, and Cys145 their contact probabilities with bisdemethoxycurcumin were increased significantly and Cys145 is a key catalytic residue which is consistent with a previous study (Jin et al., 2020). Pro168, His41, and Val116 residues contact probabilities between demethoxycurcumin-Mpro and bisdemethoxycurcumin-Mpro decreased (Table 3). Previous studies have shown COVID-19 main protease possesses a dynamic binding pocket loop comprising domain-I (residues 10–99) and domain-II (residues 100–184), in which most of the residues in this study laid (Jin et al., 2020; Kouznetsova et al., 2020; Kandeel, Kitade & Almubarak, 2020).

Figure 1 Interactions of selected polyphenols and Main protease binding pocket residues.

For all polyphenols, carbon atoms are shown in black, oxygens in red, and nitrogens in blue. Bonds in the Diferuloylmethane, Demethoxycurcumin, and Bisdemethoxycurcumin are shown in purple, and bonds in binding pocket residues are in brown. Hydrogen bonds are shown (with their lengths) as green dashed lines. Residues making hydrophobic interactions with the Diferuloylmethane, Demethoxycurcumin, and Bisdemethoxycurcumin are shown as red arcs with radiating lines. Diferuloylmethane, Demethoxycurcumin, and Bisdemethoxycurcumin atoms involved in these hydrophobic interactions are shown with radiating red lines.

Table 2 Inter-residue distances and corresponding standard deviations. In each residue pair, one belongs to the binding pocket loop region and the other belongs to surface part of the enzyme.

Residues	Free enzyme	Enzyme-demethoxycurcumin complex	Enzyme-bisdemethoxycurcumin complex	
Arg188–Phe294	5.154 ± 1.531	5.937 ± 1.604	4.5 ± 1.10	
Val202–Gln110	9.419 ± 1.440	8.455 ± 1.090	9.25 ± 1.440	
Gly109–Gln240	10.111 ± 1.838	8.296 ± 1.459	11.31 ± 1.14	
Pro293–Gln192	8.570 ± 1.663	7.095 ± 1.022	5.13 ± 1.04	
Ile249–Asn142	9.170 ± 5.015	10.485 ± 1.560	6.71 ± 0.44	
Pro252–Lys137	7.11 ± 0.02	8.5 ± 1.4	5.63 ± 1.04	
Thr292–Thr199	5.109 ± 0.440	6.039 ± 0.940	4.110 ± 0.230	
Ile200–Leu272	7.213 ± 0.018	9.216 ± 0.218	6.121 ± 0.618	

Figure 2 Plots of root-mean-square deviations of free SARS CoV-2 main protease (Mpro) (blue) and the complex of SARS CoV-2 main protease (7BUY) (red) with three polyphenols along the MD simulation time.

Free SARS CoV-2 main protease (Mpro) (blue) and the complex of SARS CoV-2 main protease (7BUY) (red) with three polyphenols.

Table 3 Residues forming contacts with diferuloylmethane polyphenol, demethoxycurcumin polyphenol and bisdemethoxycurcumin polyphenol in binary complexes.

Residues	Enzyme-demethoxycurcumin complex	Enzyme bisdemethoxycurcumin complex	Enzyme-diferuloylmethane complex	
Gln192	0.70 ± 0.01	0.85 ± 0.01	0.45 ± 0.02	
Leu167	0.90 ± 0.03	0.98 ± 0.004	0.60 ± 0.05	
Tyr239	0.60 ± 0..07	0.80 ± 0.01	0.30 ± 0.01	
Asp289	0.70 ± 0.19	0.90 ± 0.004	0.60 ± 0.1	
Arg188	0.75 ± 0.011	0.80 ± 0.05	0.73 ± 0.09	
Glu166	0.80 ± 0.04	0.80 ± 0.01	0.90 ± 0.01	
Glu288	1.00 ± 0.06	0.50 ± 0.01	0.60 ± 0.12	
His114	1.00 ± 0.00	1.00 ± 0.00	0.50 ± 0.03	
Gln189	0.70 ± 0.05	0.90 ± 0.00	1.00 ± 0.09	
Pro168	0.30 ± 0.01	1.00 ± 0.00	0.85 ± 0.05	
Gln189	0.45 ± 0.02	1.00 ± 0.00	0.40 ± 0.16	
Cys145	0.75 ± 0.09	0.90 ± 0.01	0.60 ± 0.06	
Met165	0.80 ± 0.1	0.85 ± 0.12	0.50. ± 0.12	
His 164	0.40 ± 0.00	0.70 ± 0.01	1.00 ± 0.04	
Asp187	0.90 ± 0.03	0.90 ± 0.03	0.90 ± 0.03	
Arg188	0.93 ± 0.12	0.89 ± 0.01	0.76 ± 0.23	

To examine the stability and conformational dynamics of the Mpro–three curcumin-derived polyphenols complex and free Mpro, we calculated the RMSD for the backbones of all residues (Figs. 2 and 3). In this study, much attention has been given to the impact of curcumin derived polyphenols diferuloylmethane, demethoxycurcumin, and bisdemethoxycurcumin on the structure and dynamics of the main protease (Mpro). Here free Mpro has a significantly large root-mean-square deviation compare to curcumin sp polyphenols binding the Mpro pocket domain. The time evolution plot of RMSD displays the complex structure of diferuloylmethane- Mpro, demethoxycurcumin-Mpro, and bisdemethoxycurcumin-Mpro attain equilibrium at 1,000 ps and remains stable up to 5,000 ps (Fig. 2). In particular, the complex structure with demethoxycurcumin and bisdemethoxycurcumin is largely stabilized around 3,000–6,000 ps (Fig. 3). Likewise, its fluctuation main protease complex is smaller, especially in the segment Asn226–Gly252, which is the active-site loop region which is consistent with the previous study discussed (Fig. 4). Particularly, binding of the bisdemethoxycurcumin was observed to decrease RMSF of some segments (30–100, 200–250, and 250–300) and increase that of the segment (100–150) (Fig. 4). The higher value of RMSF depicted that the structure has some flexible regions while the lower value of RMSF indicated that the structure was good in terms of secondary structure (O’Boyle et al., 2011). Therefore, the result revealed the interactive stabilizing effect of Demethoxycurcumin and Bisdemethoxycurcumin. Also, the impact of these polyphenols on the structure and dynamics of the Main protease binding pocket domain region was further regulated by a hydrogen bond. Here, we also examine the time evolution of the hydrogen bond (H-bond) formed between the Mpro and curcumin sp derived polyphenols (Fig. 5). We found that the trajectory of bisdemethoxycurcumin revealed five H-bonds with Mpro, three H-bonds were seen consistently during the simulation (Fig. 5). Demethoxycurcumin revealed the presence of ten H-bonds with Mpro but, three H-bond can be seen retained throughout the simulation, the H-bond plot of diferuloylmethane shows a maximum of seven H-bond interactions with Mpro, but only two seen retained throughout the simulation (Fig. 5), while the carmofur showed a lesser number of hydrogen bonds (Jin et al., 2020). The strong hydrogen bonding interactions between the demethoxycurcumin-Mpro and bisdemethoxycurcumin-Mpro may be a potential inhibitor (Bolton et al., 2008). This study may contribute to the understanding of the structure-based design of COVID-19 therapy with curcumin-derived polyphenols. Future investigations are necessary to validate the kinds of conclusions that can be drawn from this study.

Figure 3 Plots of Root-mean-square deviations of free main CoV-2 protease (Mpro) (gray) and the complex of Mpro with Bisdemethoxycurcumin (yellow), Demethoxycurcumin (blue) and Diferuloylmethane (red) along the MD simulation time for three individual polyphenols.

Free main CoV-2 protease (Mpro) (gray), the complex of Mpro with Bisdemethoxycurcumin (yellow), Demethoxycurcumin (blue) and Diferuloylmethane (red).

Figure 4 RMSF plot of free main CoV-2 protease (Mpro) (yellow) and the complex of Mpro with Bisdemethoxycurcumin (red), Demethoxycurcumin (blue) and Diferuloylmethane (gray) along the MD simulation time for three individual polyphenols.

Free main CoV-2 protease (Mpro) (yellow), the complex of Mpro with Bisdemethoxycurcumin (red), Demethoxycurcumin (Blue) and Diferuloylmethane (gray).

Figure 5 Number of hydrogen bond present in Bisdemethoxycurcumin-SARS-CoV-2 main protease (yellow), Demethoxycurcumin-SARS-CoV-2 main protease (gray), free SARS-CoV-2 main protease (red) and diferuloylmethane-main protease (blue).

Bisdemethoxycurcumin-SARS-CoV-2 main protease (yellow), Demethoxycurcumin-SARS-CoV-2 main protease (gray), free SARS-CoV-2 main protease (red) and diferuloylmethane-main protease (blue).

Conclusions

Until now, Great progress has been made in Computational methods involving drug target identification in a paradigm change in both industry and academics. Computational techniques, for example, data mining, homology modeling, MD simulation, cheminformatics, docking, and QSAR modeling have provided powerful techniques for target identification, drug discovery, and optimization. First, curcumin-derived polyphenols were subjected to in silico ADMETox and we found the potential ability to work effectively as drugs. Based on the results of the present study, it can be concluded that the investigated curcumin-derived polyphenols could interfere with the important residues in the enzymatic binding pocket to inhibit the main protease enzyme COVID-19 virus. Demethoxycurcumin and bisdemethoxycurcumin polyphenols are identified to have inhibitory activities against novel COVID-19 main protease. Bisdemethoxycurcumin has a stronger bond and high affinity with the Main protease (Mpro). Furthermore, the average RMSD values of the backbone atoms in docked curcumin sp derived polyphenols were calculated from 10,000 ps and showed stable RMSD values between 1 nm to 2 nm for Bisdemethoxycurcumin and Demethoxycurcumin at the reasonably consistent temperature (∼300 K) and pressure (1 bar), whereas diferuloylmethane complex showed RMSD value between 5.5 and 6.2 with same cut-off parameters. These data validated that the docked Mpro-Bisdemethoxycurcumin complexes and Mpro-Demethoxycurcumin complexes are more stable than the Mpro-diferuloylmethane complex. To completely investigate the effects of curcumin-derived polyphenols, this study recommended the roles of polyphenols identified for further exploration in a wet lab experiment.

Supplemental Information

Supplemental Information 1 Supplemental Figures and Tables.

Click here for additional data file.

Supplemental Information 2 Figures and raw data of number of hydrogen bonds, RMSD and RMSF in free protein and protein-ligand complex.

Click here for additional data file.

Supplemental Information 3 Raw data of hydrogen bonds generated during molecular dynamic simulation.

Click here for additional data file.

Supplemental Information 4 Raw data: The time series molecular dynamics simulation result of RMSD at a given time for free SARS CoV-2 main protease (Mpro) only.

The time series molecular dynamics simulation result of RMSD at a given time for free SARS CoV-2 main protease (Mpro) only.

Click here for additional data file.

Supplemental Information 5 Raw data: The time series molecular dynamics simulation result of RMSD at a given time for Demethoxycurcumin- free SARS CoV-2 main protease (Mpro) only.

Click here for additional data file.

Supplemental Information 6 Raw data: The time series molecular dynamics simulation result of RMSD at a given time for Diferuloylmethane- free SARS CoV-2 main protease (Mpro) complex only.

Click here for additional data file.

Supplemental Information 7 Raw data: The time series molecular dynamics simulation result of RMSD at a given time for Bisdemethoxycurcumin- free SARS CoV-2 main protease (Mpro) complex only.

Click here for additional data file.

Supplemental Information 8 The ADMET properties of Diferuloylmethane compounds computed using the SwissADME webserver.

Click here for additional data file.

Supplemental Information 9 The ADMET properties of Demethoxycurcumin compounds computed using the SwissADME webserver.

Click here for additional data file.

Supplemental Information 10 The ADMET properties of Bisdemethoxycurcumin compounds computed using the SwissADME webserver.

Click here for additional data file.

Supplemental Information 11 Collection of commands used to run NAMD during simulation.

Click here for additional data file.

Supplemental Information 12  

Click here for additional data file.

Supplemental Information 13 Figure 1 Interactions of selected polyphenols and Main protease binding pocket residues. For all polyphenols, carbon atoms are shown in black, oxygens in red, and nitrogens in blue. Bonds in the Diferuloylmethane, Demethoxycurcumin, and Bisdemethoxycurcum.

Click here for additional data file.

Supplemental Information 14 Figure 1 Interactions of selected polyphenols and Main protease binding pocket residues. For all polyphenols, carbon atoms are shown in black, oxygens in red, and nitrogens in blue. Bonds in the Diferuloylmethane, Demethoxycurcumin, and Bisdemethoxycurcum.

Click here for additional data file.

Supplemental Information 15 Figure 1 Interactions of selected polyphenols and Main protease binding pocket residues. For all polyphenols, carbon atoms are shown in black, oxygens in red, and nitrogens in blue. Bonds in the Diferuloylmethane, Demethoxycurcumin, and Bisdemethoxycurcum.

Click here for additional data file.

Supplemental Information 16 Figure 1 Interactions of selected polyphenols and Main protease binding pocket residues. For all polyphenols, carbon atoms are shown in black, oxygens in red, and nitrogens in blue. Bonds in the Diferuloylmethane, Demethoxycurcumin, and Bisdemethoxycurcum.

Click here for additional data file.

For the success of this work we would like to acknowledge effort of our family for their inspiration.

Additional Information and Declarations

Competing Interests

Author Contributions

Data Availability

The authors declare that they have no competing interests.

Aweke Mulu conceived and designed the experiments, performed the experiments, analyzed the data, prepared figures and/or tables, authored or reviewed drafts of the paper, ran auto-dock and NAMD simulation on his computational lab, and approved the final draft.

Mulugeta Gajaa analyzed the data, prepared figures and/or tables, authored or reviewed drafts of the paper, and approved the final draft.

Haregewoin Bezu Woldekidan analyzed the data, authored or reviewed drafts of the paper, and approved the final draft.

Jerusalem Fekadu W/mariam analyzed the data, prepared figures and/or tables, authored or reviewed drafts of the paper, and approved the final draft.

The following information was supplied regarding data availability:

Raw data is available in the Supplemental Files.

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
