# Peer review of "The impact of curcumin derived polyphenols on the structure and flexibility COVID-19 main protease binding pocket: a molecular dynamics simulation study"

_PeerJ, doi:10.7717/peerj.11590_

## Round 0.1 · original submission · Major Revisions

The manuscript got very critical remarks suggesting that I reject it in the current form. However due to the importance of the problem the journal may consider it after proper revision taking into account all the remarks and experimental validation on molecular interactions and dynamics studies. Please, follow the remarks by reviewer #2 reformatting the text. Please check recent citations, not using references to unpublished works.

Reviewer 1 ·

Basic reporting

1. INTRODUCTION (lines 5-6) and rest of the manuscript: the authors make errors in the naming of COVID-19 and SARS-CoV-2. SARS-CoV-2 is the virus that causes COVID-19 (COronaVirus Disease-19). Please make sure that this is correct! Since the paper covers the study of SARS-CoV-2 protease, the use of COVID-19 is not that appropriate. (unless you write it as COVID-19 virus)

2. The language needs to be improved considerable, particularly in the introduction and abstract. There are several errors which make it difficult to understand the manuscript. For example (lines 21-24); "Crystal structure of the CoVID-19 Mpro were obtained from the protein data bank gave a major advance for structure-based drug design of natural polyphenols and proteins gives well-established techniques to derive and to reveal important dynamic information in proteins."

3. The use of terminology is not consistent and sometimes incorrect. For example;
variations on COVID-19 found throughout the text: Covid-19, COVID_19, CoVID-19
Abbreviation of Mpro was introduced but not consistently used

4. Some of the literature cited is not yet published or still under review, hence how can this be used as a good reference? For example, references; 9, 23

5. There is no "results" section.

Experimental design

1. The research questions could be defined more clearly. This is probably due to language issues.

Validity of the findings

1. The authors state that they analyzed the binding mode of Carmfur and identified the hydrogen bonding residues. However, the seems to have already been reported in the reference that they cite along with it and has been investigated in detail in reference 7. Maybe this statement is incorrect? I don't understand

2. The authors analyze 3 polyphenols. The seem to claim that two of these had better binding modes. However, to me this is not clear from the analysis since there is not a clear difference.

·

Basic reporting

The author appraised this paper by combining docking simulation and biochemical study. However, your article is inadequately presented. Furthermore, there are many grammatical mistakes and spelling mistakes, as well. Also, the similarity index is very high (around 45%).
Although the article has scientific rigour, several major flows need to be corrected before publication.

Experimental design

1. The authors just have written several issues haphazardly. Many sentences/information throughout the manuscript have serious flaws that withdrawn my attention from it.
2. Many non-scientific and incorrect/wrong information/sentences are there, which may mislead the readers.
3. The current manuscript will add no new information/recommendation for the readers.
4. Every section of the manuscript must be written scientifically according to the published literature with appropriate references.
5. The work seems preliminary. The study problem is not apparent. Can the authors explain the importance of natural polyphenol in the introduction section?
6. The article title is very ambiguous. The authors need to simplify the title.
7. The abstract section is unsuitable—no focus point in the abstract section.
8. In the introduction section, the authors provided some information about the traditional usage of the species used and tested as anti-cancer, anti-oxidant, anti-inflammatory, antiviral and so on. It is unclear why they choose the species for antiviral drug development? It is possible for any plant species to have some effects on any disease, which is common. There is no uniqueness of this species concerning the above-mentioned activity.
9. The introduction section is inapplicable. Need to change the introduction considerably. Try to include the existing research limitations also, how the present research unravels those limits.
10. English is poor. The authors need to improve their writing style. The whole manuscript needs to be checked by native English speakers.
11. Authors are suggested to replace \'CoVID-19\' with \'COVID-19\' throughout the manuscript.
12. The author should discuss more docking simulation studies in the discussion section.
13. In the discussion part, the author claimed that their suggested compound bind in the active cavity of the targeted protein but they did not write which amino acid particularly bind with the compound.

Validity of the findings

1. The experimental section is poorly designed for further docking analysis. Why the author choose the main protease crystal structure 7BUY? They could use them to analyze the active site instead of other servers which do not have proper validation or cross-checking with other tools.
2. In the method sections, the software's they used were not explained properly—lack of ligand preparation section for docking analysis.
3. Result section is a parody. Nothing was explained although very less explanation can be found in the discussion section. Why are the active site residues important? The compounds they found how they are interacting with active site lacks explanation.
4. The study is based on Mpro protein. The Mpro protein is the most abundant protein in coronavirus, and it is normally conserved. Have the authors tried to explore in silico study in unique structural proteins such as spike protein?
5. In silico prediction is unlikely fitting with the wet-lab data. Experimental validation of the predicted compounds is highly required to recommend any predicted target as a potential vaccine.
6. The conclusion needs to address future perspectives.

Additional comments

I found this manuscript very groundwork and do not recommend to publish in the present form. I also find the present manuscript lacks for its originality. Radical changes are obligatory before go for the final version.

·

Basic reporting

The title is not appropriate for this manuscript because the authors only include 3 polyphenols in this study.

Why did the authors do the molecular dynamics in 50 nanoseconds? I consider a very short time to reach a final conclusion.
In line 228-230: two polyphenols are identified, namely curcuma sp, demethoxycurcumin and bisdemethoxycurcumin polyphenoles that have inhibitory activities against new COVID-19 main protease.
Please; corrects that turmeric sp is not a polyphenol.
Authors should have done an in-silico pharmacokinetic evaluation to understand how these molecules could cross the gastrointestinal barrier and be druggable. Many molecules are actives in docking test but they have a poor absortion by oral route and it is the main problem of turmeric supplements. Please, try to improve your discussion considering this topic in regard to the pharmacokinetic properties of curcuminoids as well as limitations of this study.

Experimental design

no comments

Validity of the findings

no comments

---

## Round 0.2 · Major Revisions

The manuscript became better after first review round. However it needs additional revision on the text presentation and in the resulting part. Please update the text according to the remarks by reviewer #1, add more results and compare current docking studies on SARS-CoV2 protein. Check cross-references on the literature and the tables in the main text. The topic is really important. Welcome to resubmit the revised version to PeerJ soon.

·

Basic reporting

The author appraised this paper by combining docking simulation and biochemical study. However, your article is inadequately presented. Furthermore, there are many grammatical and spelling/typos mistakes, as well.
Although the authors made several corrections, it still needs some additional revisions.

Experimental design

1. Still, there are many non-scientific and incorrect/wrong information/sentences that are there, which may mislead the readers. For example, lines 68, 78 etc.
2. Most data are now backdated since we are now at the end of September. I suggest to accept it after updating it with the addition of newly updated data
3. English is still low. Needs extensive language editing.
4. There are many syntax errors and awkward sentences. Please carefully edit the whole manuscript sentence by sentence to improve its readability. For example, lines 68, 78, etc.
5. There are lots of redundant sentences repeating in the whole paper, making me feel the paper like a “cut and paste” from several other resources. For example, line 77: “ Among phytochemicals compounds, polyphenols possessing a variety of potential biological benefits such as” and line 66: “ Up to now, there are no robust drugs and vaccines for wide-spread SARS-CoV-2 virus. Finding a new drug in the wet lab and bringing it to market is hard, expensive, and time-consuming”. The latter sentence is not a way of scientific description. I suggest deleting sentences like this (there are more throughout the paper).
6. The authors need to deploy the sentences more scientifically.
7. Some sentences are not standard English and feels strange.
8. The author should discuss more docking simulation studies in the discussion section.
9. In the discussion part, the author claimed that their suggested compound bind in the active cavity of the targeted protein but they did not write which amino acid mainly bind with the compound.

Validity of the findings

1. Tables in absolute numbers do not make sense.
2. In silico prediction is unlikely fitting with the wet-lab data. Experimental validation of the predicted compounds is highly required to recommend any predicted target as a potential vaccine.
3. The conclusion needs to address future perspectives.

Additional comments

I found this manuscript basis and do not recommend to publish in the present form. I also find the current manuscript lacks for its originality. Radical changes are obligatory before go for the final version. Therefore, the article in this present form cannot be published in this journal.

·

Basic reporting

After reviewing the manuscript titled "The impact of curcumin derived polyphenols on the structure and flexibility COVID-19 main protease binding pocket: A molecular dynamics simulation study" and corrected the commentaries of reviewers. I suggest to approve this manuscript.

It is necessary to improve the spelling of English grammar. Table 3 of ADME prediction in Supplementary material is absent.

Experimental design

no comment

Validity of the findings

no comment

Additional comments

no comment

---

## Round 0.3 · Minor Revisions

Thanks for the text update. The manuscript still needs some obligatory revisions before publication. Please check comments by reviewer #1. Since articles on COVID-19 and even specifically for protein structure modeling will attract public attention, I’d recommend adding citations to recent publications, including specialized conferences and recent papers at PeerJ. The topic is highly important. Check again English presentation.

Please update the text soon to avoid further delay. Waiting for the revised manuscript.

·

Basic reporting

The author appraised this paper by combining docking simulation and biochemical study. However, your article is inadequately presented. Furthermore, there are many grammatical and spelling/typos mistakes, as well.

Although the authors made several corrections, it still needs some additional minor revisions.

Experimental design

1. Still, there are many non-scientific and incorrect/wrong information/sentences are there, which may mislead the readers. For examples, lines 16, 17, 68 etc.
2. There are many syntax errors and awkward sentences. Please carefully edit the whole manuscript sentence by sentence to improve its readability. For examples, lines 72-74, etc.
3. The authors need to deploy the sentences more scientifically.
4. In abstract, authors are advised to depict the precise outcome of work. Like what are you revealed from molecular dynamics simulation of vaccine candidate and its impact?
5. Some sentences are not standard English and feels strange.
6. What was algorithm of molecular docking tool, add it, Has author performed protein- ligand profiler of vaccine, it will be helpful to define the molecular contacts? Also explain how was interactions identified.
7. Authors have employed the computational docking and MD simulation analysis but not have given proper reference.
8. Overall language and of the manuscript should be improved and correct the spelling mistakes.
9. In the discussion part, the author claimed that their suggested compound bind in the active cavity of the targeted protein but they did not write which amino acid mainly bind with the compound.
10. Can the authors compare their results matching/ correlation with the latest vaccines in the market? Can the authors add something in manuscript related to new strains/mutations reported for SARS-CoV-2?

Validity of the findings

1. In silico prediction is unlikely fitting with the wet-lab data. Experimental validation of the predicted compounds is highly required to recommend any predicted target as a potential vaccine.
2. Can the authors add something in manuscript related to new strains/mutations reported for SARS-CoV-2?
3. Why no docking figures in the main text file?

Additional comments

Though the authors addressed some of my comments but still, I found this manuscript basis and do not recommend to publish in the present form. Minor changes are obligatory before go for the final version.

·

Basic reporting

Authors proposed to investigate the inhibitory effect against COVID-19 and identification of important residues for interaction with curcumin sp derived polyphenols. Based on the results of the present study, it can be concluded that the investigated curcumin sp derived polyphenols could be interact with the important residues in the enzymatic binding pocket to inhibit the main protease enzyme Covid-19 virus. Demethoxycurcumin and Bisdemethoxycurcumin have a hydrogen bonds and high affinity with Main protease (Mpro). Furthermore, the average RMSD values of the backbone atoms in docked curcumin sp derived polyphenols were calculated from 10000ps and showed stable RMSD values between 1 to 2nm for Bisdemethoxycurcumin and Demethoxycurcumin at the reasonably consistent temperature (∼300 K) and pressure (1bar), whereas diferuloylmethane complex showed RMSD value between 3.0 to 6.0nm with same cut-off parameters. These data validated that the docked Mpro-Bisdemethoxycurcumin complexes and Mpro-Demethoxycurcumin complexes are more stable than Mpro-diferuloylmethane complex. In order to completely investigate the effects of curcumin sp derived polyphenols, this study recommended the roles of polyphenols identified for further exploration in experiments in vivo and in vitro.

Experimental design

The experimental design was improved according to my comments.

Validity of the findings

The findings were supported by the statistical analysis and several methods used during thw in-silico analysis.

Additional comments

The authors improved and corrected my comments based on my comments in previous reviews. This manuscript could be accepted in its current form.

---

## Round 0.4 · Minor Revisions

The manuscript still needs some obligatory minor changes. Please fix it according to the reviewer's remarks. Pleas update discussion part to refer to actual information on COVID-19 (May 2021)

·

Basic reporting

The author appraised this paper by combining docking simulation and biochemical study.
Although the authors made several corrections, it still needs some additional minor revisions.

Experimental design

1. Check the spelling of Covid-19 and SARS-COV2.
2. Some spelling mistakes are still in there (e.g., treate, Sp. etc)
3. The authors need to deploy the sentences more scientifically.
4. In discussion update the information. The current information is old (March 24, 2021, 20:04 GMT).
5. Overall language and of the manuscript should be improved and correct the spelling mistakes.
6. In the discussion part, the author claimed that their suggested compound bind in the active cavity of the targeted protein but they did not write which amino acid mainly bind with the compound. It should be corrected.
7. Use the full form (e.g., WHO) whenever use for the first time.

Validity of the findings

Looks okay now.

Additional comments

Although the authors made several corrections, it still needs some additional minor revisions.

---

## Round 0.5 · accepted · Accept

Thanks for the manuscript revision and all the technical updates. Finally the reviewers have no more critical comments. The research topic on COVID-19 is really important. I endorse the publication.

·

Basic reporting

The author appraised this paper by combining docking simulation and biochemical study.
The authors made the revisions as per my comments.

The revisions look good to me. The manuscript is good enough to publish in PeerJ journal.
The authors did an excellent job.

Congratulations!!!

Experimental design

The revisions look good to me.

Validity of the findings

Looks okay now.

Additional comments

The authors made the revisions accordingly. The revisions look good to me. The manuscript is good enough to publish in PeerJ journal.